# From Gene Networks to Therapeutics: A Causal Inference and Deep Learning Approach for Drug Discovery

**DOI:** 10.3390/ph18091304

**Published:** 2025-08-30

**Authors:** Sudhir Ghandikota, Anil G. Jegga

**Affiliations:** 1Division of Biomedical Informatics, Cincinnati Children’s Hospital Medical Center, Cincinnati, OH 45229, USA; sudhir.ghandikota@cchmc.org; 2Department of Pediatrics, University of Cincinnati College of Medicine, Cincinnati, OH 45267, USA

**Keywords:** drug repurposing, drug repositioning, drug discovery, deep learning, mediation analysis, network analysis, idiopathic pulmonary fibrosis

## Abstract

**Background/Objectives:** Drug discovery is a lengthy and expensive process, taking an average of 10 years and more than USD 2 billion from target discovery to drug approval. It is even more challenging in complex diseases due to disease heterogeneity and limited knowledge about the underlying mechanisms. We present a novel computational framework that integrates network analysis, statistical mediation, and deep learning to identify causal target genes and repurposable small-molecule candidates. **Methods**: We applied weighted gene co-expression network analysis (WGCNA) and bidirectional mediation analysis (causal WGCNA) to transcriptomic data from idiopathic pulmonary fibrosis (IPF) patients to identify genes causally linked to the disease phenotype. These genes were used as a phenotypic signature for deep learning-based compound screening using the DeepCE model. **Results**: Using RNA-seq data from 103 IPF patients and 103 controls, we identified seven significantly correlated modules and 145 causal genes. Five of these genes (*ITM2C, PRTFDC1, CRABP2, CPNE7,* and *NMNAT2*) were predictive of disease severity in IPF. Our compound screening identified several promising candidates, such as Telaglenastat (GLS1 inhibitor), Merestinib (MET kinase inhibitor), and Cilostazol (PDE3 inhibitor), with significant inverse correlation with the IPF-specific gene signature. **Conclusions**: This study demonstrates the utility of combining causal inference and deep learning for drug discovery. Our framework identified novel gene targets and therapeutic candidates for IPF, offering a scalable strategy for phenotype-driven drug discovery and repurposing.

## 1. Introduction

Target selection and engagement is a crucial first step in drug discovery, and a significant percentage of the clinical trial failures are linked to incorrect target selection and poor target engagement [1]. It is even more challenging in complex diseases because of the multifactorial nature and heterogeneity of these diseases; they are often driven by multiple genes, biological pathways, and environmental factors. The interconnected nature of these biological mechanisms makes it even more challenging to identify robust druggable targets. Additionally, disease progression can alter the relevance and importance of the selected target over time, making the target selection step even more critical in complex diseases.

Conventional drug discovery approaches are mostly based on a one-drug–one-gene paradigm [2,3], where compound screening is performed against a single, specific target [4]. Understandably, these approaches have limited success in complex diseases [3], as target-based screening methods often yield a poor correlation between modulating a single protein by a chemical compound and organism-level responses [2]. Furthermore, compound screening using single-gene targets often fails to recognize off-target interactions that can lead to unintended side effects [5]. On the other hand, phenotype-based compound screening represents a systematic approach by possibly capturing the full spectrum of a compound’s biological activity [6]. They often begin with a phenotype-specific gene signature, which is queried against a reference database of small-molecule perturbations, typically using correlation measures. Moreover, deep learning (DL)-based frameworks can help mitigate the typical issues of noise and incompleteness associated with large-scale transcriptomics data [6].

We present a bioinformatic methodology that combines network analysis, statistical mediation models, and DL-based small-molecule compound screening frameworks for therapeutic drug discovery (See Materials and Methods). Our methodology starts with analyzing one (or more) transcriptomic datasets, converting them into gene co-expression networks, and applying a network analysis framework [7,8] to identify significantly correlated gene modules. Next, we employ statistical mediation techniques to examine the relationship between the phenotype and the candidate modules and identify candidate causal genes [9]. Instead of modeling the direct gene-phenotype associations, our approach enables us to capture high-level relationships between genes, network modules, and the phenotype, while adjusting for any clinical confounders. Finally, we use the significant mediator genes as the phenotypic signature and apply a mechanism-driven compound screening method to identify potential therapeutic candidates that can modulate this causal signature [2] (Figure 1). Compared to a list of dysregulated genes that are often used as the input signature in connectivity mapping approaches, causal genes make for more effective drug targets [10] as they represent high-level biological mechanisms driving phenotype progression. In contrast, differentially expressed gene (DEG) sets reflect downstream responses. The causal regulators provide potentially actionable insights for drug discovery.

To demonstrate the utility of our methodology, we used idiopathic pulmonary fibrosis (IPF) as a disease case study. IPF is a severe fibrotic lung disease characterized by chronic, progressive scarring, and destruction of the lung parenchyma, often requiring a lung transplant in advanced cases [11]. Two FDA-approved antifibrotic drugs, nintedanib and pirfenidone, are currently used as the first-line therapy for IPF. Even though both of them are effective in managing the symptoms of lung function decline in IPF patients, these drugs do not cure the disease and do not reverse the lung fibrosis [12,13]. Owing to the complexity, heterogeneity, and multifactorial nature of IPF pathogenesis, combination therapies are hypothesized to be more effective than monotherapies [14].

In this study, we applied our methodology and identified several novel candidate causal genes potentially important in the progression of IPF. Several of these were found to be significantly correlated with the forced vital capacity (FVC) and diffusing capacity of lungs for carbon monoxide (DLCO) lung function traits. Furthermore, we ran machine learning models to evaluate whether these mediator genes can predict the phenotype status using independent study cohorts. Finally, we ran a DL-based compound screening method to identify small-molecule candidates that can potentially modulate the IPF causal signature.

## 2. Results

### 2.1. Transcriptomic Datasets

In our IPF case study, we implemented a network analysis framework on a gene co-expression network, layered with a statistical mediation analysis. For constructing the gene network, we used RNA-seq data from whole lung tissues of 103 IPF samples and 103 healthy controls [15]. Additionally, we used two other IPF RNA-seq datasets, GSE124685 [16] and GSE213001 [17], to validate the biomarker candidates identified in the network analysis step (Table 1). All three publicly available datasets were downloaded from NCBI’s Gene Expression Omnibus (GEO) repository [18]. For compound screening, we used the Library of Integrated Network-Based Cellular Signatures (LINCS) [19], a small-molecule gene expression database of drug perturbation profiles.

### 2.2. Correlated Gene Modules

Gene expression profiles from whole lung tissues (GSE150910) from both IPF and control samples were used for constructing gene co-expression networks and running the WGCNA algorithm [7]. Before this, we normalized the RNA-seq counts using the *voom* (variance modeling at the observational level) method [20]. We identified sixteen non-overlapping gene modules (Figure 2), out of which seven modules were significantly (Student asymptotic *p*-value < 0.05) correlated/upregulated in IPF (Appendix A). The *greenyellow* module (486 genes) was the most significant module (Figure 2) that contained genes predominantly involved in extracellular matrix organization (ECM) (GO:0030198) and collagen fibril organization (GO:0030199). Furthermore, five out of the seven correlated modules (*greenyellow, brown, yellow, red,* and *purple)* were strongly associated with lung function (FVC and DLCO) (Figure 2). Finally, we used these seven significantly correlated modules to run the mediation analysis to identify potential candidate causal genes.

### 2.3. Candidate Causal Genes in IPF

We implemented mediation analysis among the phenotype, the module, and the module genes using the CWGCNA framework [9]. Before this, using type-III ANOVA models, we tested for any potential confounding effects of several different clinical variables, including age, gender, and smoking status, and all mediation analysis models were adjusted for statistically significant (age and smoking status) effects (See Materials and Methods). We applied bidirectional mediation models for each of the seven candidate WGCNA modules and identified 145 unique mediator genes with significant mediation that are potentially drivers (forward direction) of the disease progression in IPF (Appendix A). The majority of them (114 out of 145 genes) were found to be significantly upregulated (${log}_{2}FC>0.58$ and adjusted *p*-value < 0.05) in IPF (Figure 3). Similarly, 101 out of the 145 genes were significantly associated (adjusted *p*-value < 0.05) with both FVC and DLCO lung function traits (Appendix A). We also found that 35/145 genes are part of the druggable genome collection, indicating their potential as therapeutic targets. This gene list represents the set of targets that are known or predicted to interact with drug compounds and was manually compiled from multiple sources, including DrugBank [21], DGIDb (v5.0) [22], and the CLUE-Drug repurposing hub [23].

Furthermore, we identified a significant (*p*-value < 0.05) number of mediator genes (37 out of 145 genes) that were previously known to be associated with the IPF phenotype. To do this, we compiled a list of known IPF-associated genes (EFO_0000768; 2910 genes) from the Open Targets Platform that mines and scores disease target genes from multiple evidence data sources, including GWAS associations, literature curation, and clinical evidence [24] (Appendix A). Additionally, we found several IPF lung single-cell markers (RAD51, CDKN3, TROAP, TAFA3, SGO1, NCAPH, CEP55, UTS2, ANLN, BUB1B, DGKI, and SKA3) among the remaining 108 causal genes from our mediation models.

### 2.4. Candidate Genes Enriched in Pro-Fibrotic Niches

To identify potential disease-driving mediator candidates localized in the lung tissue, we utilized a spatial transcriptomics study in IPF [25]. In this study, Mayr et al. identified three disease-associated niches (the fibrotic niche, the airway macrophage niche, and the immune niche) with unique cellular compositions and localizations. Intersecting the niche markers (${log}_{2}FC>0.25$ and adjusted *p*-value < 0.05) with the significant mediator candidates, we identified four genes (CRABP2, MKI67, PRDX4, PLPP5) found in all three IPF niches (Appendix A). In one of our earlier studies, we demonstrated that CRABP2, a retinol-binding protein, is dysregulated in multiple lung cells of IPF and is strongly correlated with decline in lung function [26]. In that study, immunostaining of IPF lung sections demonstrated increased levels of CRABP2 in both spindle-shaped fibroblasts and airway epithelial cell types in the distal fibrotic lung lesions of IPF compared to control lungs. Similarly, studies analyzing human serum samples obtained from IPF patients have shown that PRDX4 (peroxiredoxin-4) is associated with the aggravation of inflammatory changes and fibrotic effects in the pathogenesis of IPF [27]. Furthermore, elevated levels of serum PRDX4 were associated with poor prognosis in murine models, demonstrating its potential clinical significance.

Additionally, we identified 24 genes (*p*-value < 0.05) belonging to the fibrotic niche, consisting of myofibroblasts and aberrant basaloid cells, located around the airways [25]. Similarly, we identified 20 genes (not statistically significant; *p*-value = 0.3) from the airway macrophage niche in the lumen containing SPP1^+^ (Osteopontin) macrophages (Appendix A). Our mediator gene list also included 36 genes (*p*-value < 0.05) from the immune niche within IPF tissue, predominantly characterized by lymphoid cells surrounded by remodeled endothelial cells. In total, our mediation analysis identified 54 unique candidates that are marker genes in at least one of the three pro-fibrotic niches (Appendix A).

### 2.5. Mediator Genes Associated with IPF Severity

Next, to identify candidate causal genes associated with severe IPF phenotypes, we intersected the significant mediator candidates with the upregulated genes (${log}_{2}FC>0.58$ and adjusted *p*-value < 0.05) in progressive-stage and end-stage IPF tissue from a lung RNA-seq dataset (GSE124685) [16]. The study used quantitative micro-CT imaging and tissue histology on IPF lung samples to categorize three distinct stages of fibrotic remodeling (early-stage or IPF1, progressive-stage or IPF2, and end-stage or IPF3). We identified 40 mediator genes (*p*-value = 4.69 × 10^−13^) that were upregulated in end-stage IPF and 38 candidate genes (*p*-value = 8.07 × 10^−14^) overexpressed in progressive-stage IPF samples. There were 30 mediator genes upregulated in both IPF2 and IPF3 samples. Additionally, we also found 40 candidate genes (*p*-value = 2.88 × 10^−16^) that were upregulated in early-stage IPF (Appendix A).

### 2.6. Biomarker Candidates in IPF

To assess the predictive ability of the mediator candidate genes and identify potential biomarker candidates in IPF, we ran generalized linear models (GLMs) via penalized maximum likelihood [28,29]. We used an independent IPF cohort (Table 1) consisting of 61 IPF cases (including 22 advanced and 27 severe cases) and 40 control samples to train the GLM models. These models were evaluated using the area under the precision-recall curves (AUPRC) and receiver operating characteristic (AUROC) curves (See Materials and Methods). We identified several candidate genes (ITM2C, PRTFDC1, CRABP2, CPNE7, FAM83D, NMNAT2, P4HA3, PDGFD, and PAPP2) with high area under the curve (AUC > 0.9) scores (Table 2). Among them, the integral membrane protein 2C (ITM2C) gene performed the best with high AUPRC and AUROC scores (Figure 4). The ITM2C gene is a known marker of plasmablast cells in the lung [30], which are implicated in the progression of IPF [31].

We also trained GLM models to distinguish severe and advanced IPF samples from controls (Figure 4). Based on the AUC scores, we identified seven gene candidates (ITM2C, CPNE7, PRTFDC1, CRABP2, NMNAT2, LAX1, and PAPPA2) with strong predictive performance in classifying severe IPF samples (Table 2). Similarly, we found nine additional gene candidates (ITM2C, CRABP2, PRTFDC1, FAM83D, NMNAT2, MYOF, P4HA3, CDH3, and CPNE7) highly capable of distinguishing advanced IPF samples from the healthy controls (Table 2 and Appendix A). Five of the mediator candidates (ITM2C, PRTFDC1, CRABP2, CPNE7, and NMNAT2) were strong predictive biomarkers across both sets of models (severe and advanced).

### 2.7. Small-Molecule Compounds Targeting the Causal Genes

Finally, we used a modified version of the DeepCE [2] model (See Materials and Methods) to identify compounds targeting the potentially causal genes in IPF. Specifically, we used 145 significant mediator genes (Section 2.3) as the query signature and implemented compound screening by computing Spearman’s rank-order correlation between the drug perturbation gene expression profiles and the IPF-specific signature. We identified 126 small-molecule compounds with significant negative correlation scores (*p*-value < 0.05) in any of the seven different cell lines used in the model (Appendix A). Of these, 23 compounds (Table 3) exhibited significant negative correlations in multiple (two or more) cell lines.

Among them, the compound Telaglenastat (CB-839), a glutaminase 1 (GLS1) inhibitor, was significantly negatively correlated with our causal signature in five out of the seven cell lines. This compound has been shown to be therapeutically efficacious in treating both bleomycin- and transforming growth factor-β1-induced pulmonary fibrosis in mice [32]. GLS1 inhibition is also shown to regulate collagen production in human lung myofibroblasts by promoting its stabilization [32,33]. Collagen production plays a central role in the pathogenesis of IPF with excessive collagen deposition leading to lung tissue stiffening and impaired gas exchange. Similarly, the kinase inhibitor Merestinib targets the tyrosine kinase receptor MET (hepatocyte growth factor) and is being investigated as a potential anti-cancer therapy candidate [34]. The MET-signaling pathway has been implicated in the epithelial-mesenchymal transition (EMT) and fibroblast activation in IPF [35,36]. Consequently, MET has been proposed as a functional marker and an actionable target in IPF [37]. Our analysis also highlighted Cilostazol, a Phosphodiesterase 3 (PDE 3) inhibitor approved for use as an antiplatelet agent and vasodilator, for the symptomatic relief of intermittent claudication. Other studies have demonstrated its antifibrotic effects in a rat model of liver fibrosis [38] and cardiac fibrosis in mice [39]. We also identified the ROCK inhibitor Fasudil, which is known to ameliorate experimental pulmonary fibrosis by inhibiting mechanosensitive signaling in myofibroblasts [40] (Appendix A).

### 2.8. Causal Candidate Genes and Small-Molecule Targets

To understand the potential mechanism of actions (MOAs) associated with some of the top small-molecule compound hits explicitly to IPF, we computed functional enrichment analysis by combining the 145 mediator candidate genes and drug targets (Figure 5). Specifically, we obtained targets genes for Cilostazol (68 genes), Telaglenastat (5 genes), and Merestinib (16 genes) from the PubChem [41] database (Appendix A). In the combined gene set several IPF-related processes were enriched including fibroblast proliferation (12 genes; FDR B&H = 7.03 × 10^−6^), fibroblast apoptotic process (7 genes; FDR B&H = 3.32 × 10^−6^), and extracellular matrix organization (17 genes; FDR B&H = 2.82 × 10^−5^). Additionally, we also observed enrichment of human lung cell type marker genes [42] from proliferating basal cells (21 genes; FDR B&H = 2.14 × 10^−3^), adventitial fibroblasts (10 genes; FDR B&H = 8.49 × 10^−3^), and differentiating basal cells (7 genes; FDR B&H = 0.011). Also, we found 10 genes (FDR B&H = 4.56 × 10^−8^) associated with IPF from the DisGeNET database [43].

Furthermore, to illustrate the connectivity (protein–protein interactions—PPIs) between the mediator genes and the small-molecule targets, we also constructed a joint interactome (Appendix A). In total, we found 67 mediator candidates directly interacting with at least one drug target gene. We also observed a significant number of interactions between the targets genes of the three small-molecule compounds (Appendix A). We used the ToppGene tool suite (https://toppgene.cchmc.org/; last accessed on 25 August 2025) [44] to run the functional enrichment analysis and to obtain the interaction network (Appendix A). The network visualization was carried out using the Cytoscape application (v3.10.3) [45].

## 3. Discussion

In this study, we presented a therapeutic target and compound discovery approach by combining unsupervised network analysis, statistical mediation analysis, and Deep Learning (DL) models. We applied this framework to IPF and identified 16 non-overlapping gene modules, including 7 candidate modules significantly upregulated in the phenotype. From these modules, we identified 145 mediator gene candidates with significant mediation effects, after adjusting for confounders. These genes are hypothesized to *cause* the sample group differences by being involved in high-level pathways or biological processes (modules) [9]. Indeed, several of them were found to be localized in pro-fibrotic niches in the lung tissue. We also identified 35 known drug targets among these mediator genes. Additionally, machine learning models using these mediator genes demonstrated their predictive ability to categorize IPF samples from healthy controls and also for classifying severe and advanced IPF cases from controls with strong performance metrics.

The significant mediator genes were used as the phenotype signature for compound screening to identify small-molecule candidates that can potentially reverse the IPF phenotypic effects. By using Spearman rank-order correlations between the IPF signature and drug perturbation profiles, we identified more than 120 compounds with significant negative correlations with the IPF signature across different cell lines. Several of our identified candidate drugs have been reported to have therapeutic (antifibrotic) effects in in vitro and animal models. Functional enrichment analysis has shown several IPF-relevant pathways and processes shared between the causal mediator genes and the targets of the top hit small molecules. Further validation studies centered around some of the other small-molecule candidates are warranted to identify new therapeutic strategies for attenuating or reversing lung fibrosis.

Phenotype-based compound screening is generally considered systematic and comprehensive as it utilizes drug-induced gene expression profiles as readouts [2]. In comparison to previous methods, our methodology has the advantage of combining network analysis with causal inference, using bidirectional mediation models, to generate phenotype-specific signatures. This gives us the ability to potentially encode high-level causal mechanisms in these signatures. Consequently, the screened molecules are hypothesized to be safe and efficacious. Moreover, we used a customized DL model [2] for predicting robust drug perturbation profiles across multiple cell lines from otherwise noisy expression data (See Materials and Methods). Our customized model offers higher coverage by predicting expression profiles in the best inferred genes (BING) space instead of just the 978 landmark genes [19]. This enables capturing more comprehensive and relevant phenotypic signatures for a robust compound screening process. Our framework, including the experiments in the IPF use case, can be freely accessed from https://doi.org/10.5281/zenodo.16935145, accessed on 25 August 2025.

Our approach does hold certain limitations. Firstly, our mechanism-driven compound screening approach is highly dependent on the input signature and the quality of the gene expression profiles used in the analysis. Irrelevant target genes within the signature can lead to false-positive small-molecule hits. Secondly, our mediation models assume linear relationships between the phenotype and the modules and therefore cannot identify non-linear effects. Also, unmeasured confounding effects can impact the statistical mediation models, which in turn can influence the causal phenotype signatures used in the compound screening step. Hence, proper attention should be given to test and adjust for all possible confounding variables during the mediation analysis step. Additionally, noisy and unreliable perturbation profiles can lead to unsuccessful compound hits. To manage this issue, we utilized a DL model that employs an effective data augmentation technique, which can extract useful information from noisy or unreliable expression profiles [2]. Finally, the LINCS data is based on cancer cell lines and may not accurately represent the physiology of non-cancerous tissues. Therefore, additional targeted validation experiments with relevant model systems are required to test the therapeutic hypotheses generated from implementing our framework.

As a future direction, we plan to develop compound screening approaches that can leverage single-cell expression profiles and spatially resolved cellular niches towards drug discovery applications. Using such frameworks, we can also include cell–cell interaction information to potentially capture and adjust for off-target activity of the identified drug compounds. With the help of interpretable DL models, non-linear relationships between drugs, targets, and phenotypes can be efficiently captured and better understood.

## 4. Materials and Methods

### 4.1. WGCNA and Candidate Correlated Modules

The weighted gene co-expression network analysis (WGCNA) framework is useful for identifying modules of highly co-expressed genes. Briefly, WGCNA uses pairwise Pearson correlations for constructing gene networks and converts them into topological overlap matrices that encode relative gene interconnectedness within the network [8]. Then, average linkage hierarchical clustering with adaptive branch pruning is implemented to identify modules of strongly co-expressed genes (Figure 2B) [46]. We used signed co-expression networks with the power parameter (β=8) chosen based on the scale-free topology criterion (Figure 2A) [8]. Additionally, the cluster sensitivity parameter (deepSplit) was set to the default value of 2 to achieve balanced clusters with a minimum module size set to 100 genes. Then, module eigengenes were identified to compute the module-level correlations with the phenotype status and any available clinical variables. These eigengenes are generally considered to be a module representative summarizing the expression patterns within a module and are identified through principal component analysis (PCA). Additionally, to assess the significance of the module-trait correlations, we calculated the Student asymptotic *p*-values for each correlation distribution (Figure 2C). Candidate modules were filtered using a significance threshold of *p*-value < 0.05.

### 4.2. Mediation Analysis and Candidate Causal Genes

We implemented statistical mediation analysis to reveal the causal relationships between the modules and the phenotype. Specifically, we used the CWGCNA package [9] to construct mediation models using the WGCNA module, the phenotype status, and the module genes. CWGCNA uses bidirectional models to test two causal directions: (a) the forward direction of the module driving the phenotype via the gene (module → gene → phenotype), and (b) the reverse direction of phenotype causing the module changes via the gene (phenotype → gene → module). These tests, conducted using the inverse probability weighting (IPW) method [47], help clarify the cause and the mediators of a given phenotypic condition. In both models, the module genes are used as the potential mediator and tested for their mediating effect. For each gene, the proportion mediated was computed by combining both the natural and the indirect effects, and 95% confidence intervals (CIs) were estimated via bootstrapping (100 times). Next, consistent mediator genes were chosen based on the significance and magnitude of the direct and indirect effect terms. Additionally, CWGCNA applied a bagging-SMOTE (synthetic minority oversampling technique) framework to prevent the sample size differences from influencing the mediation analysis step [9].

We applied mediation analysis models using the seven modules that showed positive correlations with the IPF phenotype (Figure 3). Before this, type-III ANOVA analysis was implemented to test for any significant confounding effects. Both gender and smoking status were determined to be significant confounding variables for the phenotype status. Next, we tested the mediation effects of each module gene from the seven correlated modules, in both the forward and reverse directions. In total, we identified 145 genes with significant mediation effects uniquely in the forward direction (causal to phenotype). These genes were used in all the downstream analyses, including the phenotype-based compound screening step.

### 4.3. Machine Learning Models for Biomarker Analysis

For identifying a subset of biomarker candidate genes, we used generalized linear models (GLMs) that were trained with the elastic net penalty [28]. These models use the individual gene expression profiles as the input and are trained to predict the sample phenotype status. We implemented a leave-one-out cross-validation (LOOCV) approach to perform parameter tuning. In each training experiment, 70% of all samples were randomly picked for training the model and performing parameter tuning, and the remaining 30% of the test samples were used for model evaluation. The area under the curve (AUC) values of both precision-recall curves and the receiver operating characteristic (ROC) curves were computed to quantify the performance of these trained models. Furthermore, we conducted 100 distinct randomized experiments with separate training and validation data splits to compute robust evaluation metrics by averaging them across all the runs. Finally, in our IPF case study, we followed the same framework for models in both the training dataset (GSE150910) and the independent transcriptomic cohort (GSE213001).

### 4.4. DeepCE Model for Small-Molecule Screening

For screening small-molecule candidates, we followed the mechanism-driven approach used in [2]. Specifically, we used a modified version of the DeepCE model trained to predict the small-molecule perturbation profiles. It uses a graph neural network (GNN) to model the chemical sub-structure information [48] encoded in the form of a compound graph. Additionally, DeepCE also employs a network embedding model [49] for encoding gene–gene interaction networks [50]. It is trained using the expression profiles from the level 5 LINCS L1000 dataset (based on 978 landmark genes) [19]. DeepCE also includes experimental information, such as the cell line information and the drug dosage, in the model training step. Furthermore, a data augmentation technique is used, wherein a high-quality training set is first generated to train the model. The trained model is then used to predict and filter the gene expression profiles associated with the unreliable experiments in the L1000 data [2].

To increase the coverage of the small-molecule perturbation profiles, we trained a new DeepCE model using the level 5 L1000 data of 10,174 BING genes (978 landmark + 9196 well-inferred genes with high fidelity). We reused the original hyperparameter values and followed the same procedure to train the new model by selecting the gene expression profiles from the seven most popular cell lines (MCF7, A375, HT29, PC3, HA1E, YAPC, HELA) and the six most popular dosages (0.04 μm, 0.12 μm, 0.37 μm, 1.11 μm, 3.33 μm, 10.0 μm). We reused the high-quality signatures to train the new model and predicted the perturbation levels of approximately 2000 small molecules and nearly 6000 compounds from the BROAD Drug Repurposing Hub [23] across all seven cell lines. The model training was conducted on a HP Z4 workstation with Intel(R) Xeon(R) W-2133 CPU 3.60 GHz, 64 GB RAM, and Nvidia Quadro RTX 8000 GPU. For screening the candidate compounds, we use Spearman rank-order correlations between the causal phenotype signature (Section 4.2) and the predicted perturbation profiles. We also computed the p-values for these correlations using a hypothesis test whose null hypothesis is that the drug and the signature have no ordinal correlation. Finally, compounds that demonstrated significant negative correlations with the phenotype signature across multiple cell lines were shortlisted as likely drug repurposing candidates.

## Figures and Tables

**Figure 1 pharmaceuticals-18-01304-f001:**
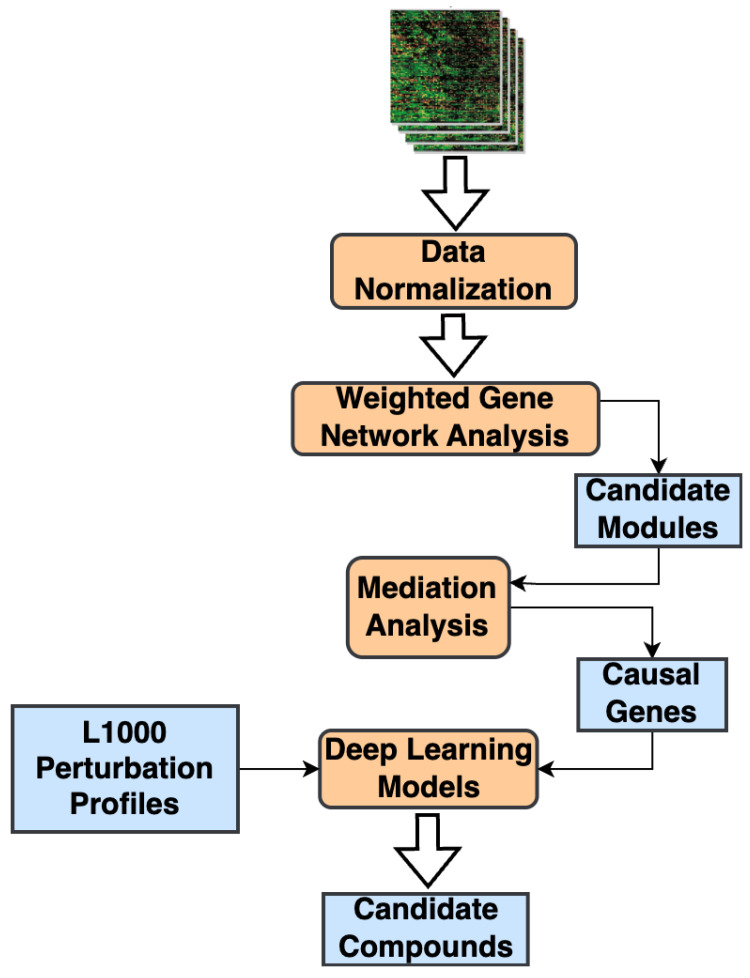
Overall workflow of our proposed compound screening methodology. Starting with a matrix of gene expression profiles as the input, the network analysis step identifies all gene modules co-expressed with respect to a phenotype. From the modules significantly associated with a phenotype, causal genes are identified by implementing statistical mediation analysis models. Finally, a multimodal deep learning model is used to identify candidate small-molecule compounds that potentially target the causal genes.

**Figure 2 pharmaceuticals-18-01304-f002:**
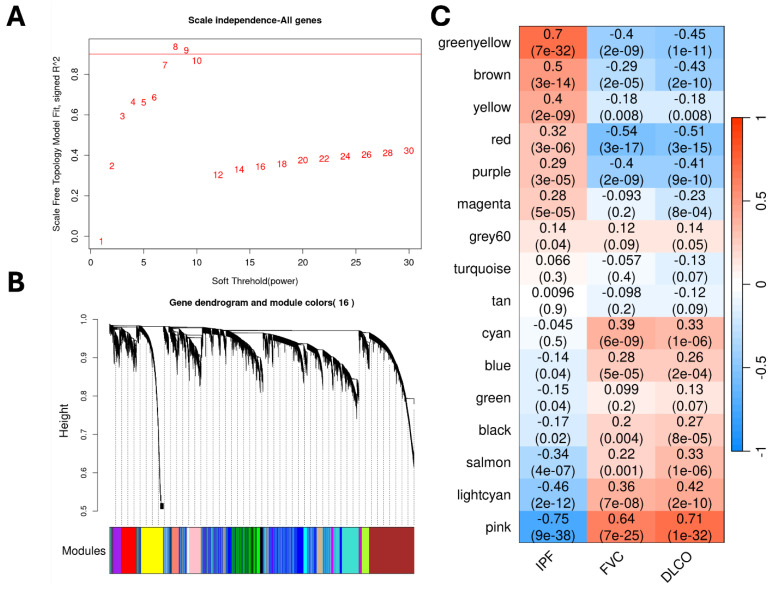
**WGCNA Steps:** (**A**) Scatter plot of the WGCNA scale-free analysis experiments on the IPF transcriptomic data. (**B**) Hierarchical dendrogram of gene modules constructed based on the topological overlap matrix (TOM). (**C**) Correlation heatmap of gene modules with the phenotypic status and the lung-function traits (FVC and DLCO) from the transcriptomic cohort.

**Figure 3 pharmaceuticals-18-01304-f003:**
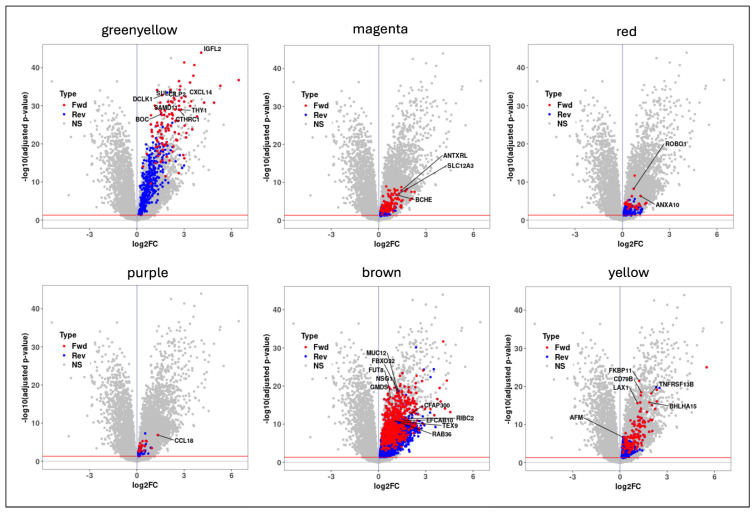
Volcano plots of the six candidate modules with significant mediator genes (red—forward; blue—reverse; gray—nonsignificant) highlighted in each of them.

**Figure 4 pharmaceuticals-18-01304-f004:**
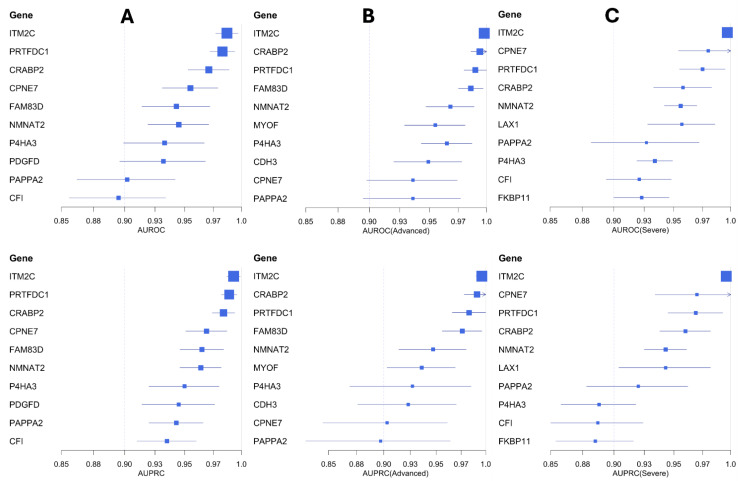
Forest plots of the top-performing biomarker candidates in classifying IPF samples. Panel (**A**) highlights the top biomarkers in classifying all IPF samples from the validation cohort (GSE213001) while Panels (**B**,**C**) include the best-performing biomarkers in categorizing advanced IPF and severe IPF samples, respectively. These biomarker genes are filtered based on the AUPRC (**bottom row**) and AUROC (**top row**) scores.

**Figure 5 pharmaceuticals-18-01304-f005:**
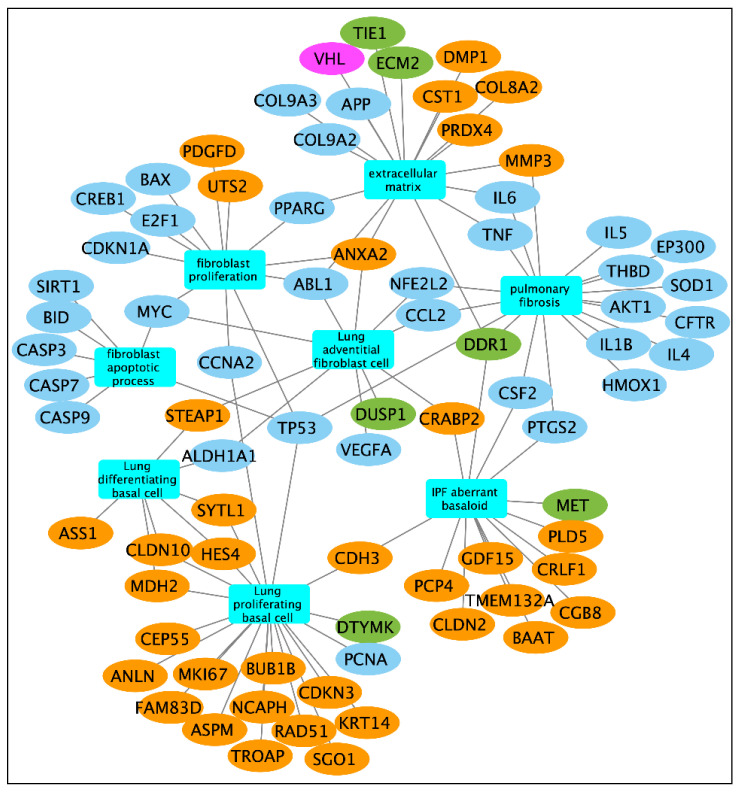
Functional enrichment network of significant mediator genes and drug targets of top hit small molecules. The mediator genes are represented as yellow elliptical nodes while the drug targets are represented in blue, green, or magenta colors. Targets genes of Cilostazol are highlighted in blue, while the green and magenta nodes represent the target genes of Merestinib and Telaglenastat, respectively. Enriched functional terms relevant to IPF are shown in turquoise-colored rectangular nodes. The network was generated using Cytoscape application (v3.10.3) and the functional enrichment was carried out using ToppFun application of the ToppGene Suite (https://toppgene.cchmc.org/; last accessed on 25 August 2025).

**Table 1 pharmaceuticals-18-01304-t001:** List of IPF transcriptomic datasets used.

GEO Accession ID	# IPF	# Controls	Reference
GSE150910	103	103	[15]
GSE124685	49 (19 = mild (IPF1); 16 = moderate (IPF2); 14 = advanced (IPF3))	35	[16]
GSE213001	61 (22 = Advanced, 27 = Severe)	40	[17]

**Table 2 pharmaceuticals-18-01304-t002:** Best-performing causal mediator genes in categorizing the IPF samples (including severe and advanced IPF) from healthy controls in an independent validation cohort (GSE213001). The table lists the AUPRC and AUROC scores averaged across 100 different experiments (both mean and standard deviation). The full list of AUC scores can be found in Appendix A.

	Gene	AUPRC	AUROC
**IPF vs. Controls**	ITM2C	0.993 ± 0.006	0.987 ± 0.01
	PRTFDC1	0.989 ± 0.007	0.983 ± 0.011
	CRABP2	0.984 ± 0.01	0.971 ± 0.018
	CPNE7	0.969 ± 0.018	0.955 ± 0.024
	FAM83D	0.965 ± 0.018	0.943 ± 0.029
	NMNAT2	0.964 ± 0.006	0.945 ± 0.026
	P4HA3	0.95 ± 0.03	0.933 ± 0.034
	PDGFD	0.945 ± 0.031	0.932 ± 0.036
	PAPPA2	0.943 ± 0.023	0.902 ± 0.04
**Severe IPF vs. Controls**	ITM2C	0.996 ± 0.004	0.997 ± 0.003
	CPNE7	0.97 ± 0.036	0.98 ± 0.026
	PRTFDC1	0.969 ± 0.024	0.975 ± 0.02
	CRABP2	0.96 ± 0.022	0.958 ± 0.025
	NMNAT2	0.943 ± 0.018	0.956 ± 0.014
	LAX1	0.943 ± 0.039	0.957 ± 0.029
	PAPPA2	0.92 ± 0.042	0.927 ± 0.045
**Advanced IPF vs. Controls**	ITM2C	0.996 ± 0.006	0.998 ± 0.004
	CRABP2	0.991 ± 0.013	0.994 ± 0.008
	PRTFDC1	0.983 ± 0.017	0.99 ± 0.01
	FAM83D	0.976 ± 0.02	0.986 ± 0.011
	NMNAT2	0.947 ± 0.033	0.968 ± 0.021
	MYOF	0.936 ± 0.033	0.955 ± 0.026
	P4HA3	0.927 ± 0.058	0.965 ± 0.022
	CDH3	0.923 ± 0.047	0.949 ± 0.029
	CPNE7	0.903 ± 0.058	0.936 ± 0.038

**Table 3 pharmaceuticals-18-01304-t003:** List of top compounds with significant negative correlation (Spearman rank-order) with the causal IPF signature. The correlations in each of the seven distinct cell lines, along with the significance *p*-values, are listed here (statistically significant correlations are highlighted in bold). Appendix A has the full list of drugs.

Drug	A375	HA1E	HELA	HT29	MCF7	PC3	YAPC
1,4-Bis((3,4-dimethoxyphenyl)sulfonyl)-1,4-diazepane	−0.25 (0.051)	**−** **0.27 (0.036)**	**−** **0.34 (0.006)**	**−** **0.28 (0.027)**	**−** **0.33 (0.009)**	−0.25 (0.053)	**−** **0.28 (0.03)**
CB-839 (Telaglenastat)	**−** **0.29 (0.023)**	0.04 (0.76)	**−** **0.29 (0.02)**	**−** **0.26 (0.044)**	**−** **0.25 (0.049)**	**−** **0.28 (0.03)**	−0.18 (0.16)
[2-(4-Amino-1,2,5-oxadiazol-3-yl)-1-ethylimidazo[4,5-c]pyridin-7-yl]-[(3S)-3-aminopyrrolidin-1-yl]methanone	−**0.31 (0.014)**	−0.18 (0.17)	**−** **0.26 (0.041)**	**−** **0.32 (0.014)**	**−** **0.31 (0.015)**	**−** **0.29 (0.02)**	−0.19 (0.13)
Aminofurazanyl-azabenzimidazole 6n	**−** **0.35 (0.005)**	**−** **0.27 (0.03)**	**−** **0.3 (0.02)**	−0.23 (0.07)	**−** **0.29 (0.023)**	−0.25 (0.053)	−0.19 (0.15)
[2-(4-Amino-furazan-3-yl)-1-ethyl-1H-imidazo[4,5-c]pyridin-7-ylmethyl]-piperidin-4-yl-amine	**−** **0.28 (0.025)**	−0.1 (0.42)	−0.25 (0.052)	**−** **0.3 (0.02)**	**−** **0.28 (0.026)**	**−** **0.3 (0.02)**	−0.09 (0.51)
Pentamidine	**−** **0.31 (0.015)**	−0.16 (0.24)	−0.21 (0.098)	−0.21 (0.11)	**−** **0.26 (0.046)**	**−** **0.26 (0.044)**	**−** **0.27 (0.034)**
RHC-80267	−0.22 (0.089)	0.08 (0.52)	−0.18 (0.16)	**−** **0.35 (0.006)**	**−** **0.29 (0.023)**	−0.23 (0.075)	**−** **0.26 (0.039)**
RK-682	**−** **0.27 (0.033)**	0.12 (0.35)	0.019 (0.88)	0.22 (0.097)	−0.18 (0.17)	−0.22 (0.091)	**−** **0.29 (0.025)**
Merestinib	−0.08 (0.54)	−0.09 (0.48)	**−** **0.33 (0.008)**	−0.06 (0.64)	**−** **0.28 (0.03)**	−0.07 (0.57)	−0.05 (0.73)
LY-255283	−0.25 (0.053)	0.17 (0.2)	−0.18 (0.16)	**−** **0.26 (0.047)**	−0.2 (0.12)	−**0.28 (0.028)**	−0.07 (0.58)
Cilostazol	**−** **0.31 (0.015)**	0.018 (0.89)	−0.15 (0.25)	−0.16 (0.22)	−0.11 (0.41)	−0.06 (0.67)	**−** **0.28 (0.033)**
M2-PK-activator	−0.19 (0.15)	**−** **0.26 (0.04)**	−0.21 (0.11)	0.25 (0.05)	−0.22 (0.089)	−0.23 (0.07)	**−** **0.29 (0.02)**
NNC-711	−0.11 (0.4)	−0.03 (0.79)	−0.2 (0.11)	**−** **0.3 (0.018)**	**−** **0.27 (0.038)**	0.1 (0.44)	0.009 (0.95)
CID 11973736	−0.23 (0.077)	−0.2 (0.12)	**−** **0.33 (0.009)**	−0.24 (0.06)	**−** **0.32 (0.014)**	−0.18 (0.16)	−0.07 (0.6)
Azeloyl diethyl salicylate	−0.15 (0.26)	0.1 (0.45)	**−** **0.34 (0.008)**	**−** **0.32 (0.011)**	−0.16 (0.22)	−0.19 (0.13)	0.02 (0.88)
Cetrimonium	**−** **0.31 (0.015)**	−0.04 (0.78)	−0.15 (0.26)	0.23 (0.07)	−0.2 (0.13)	**−** **0.28 (0.027)**	−0.24 (0.06)
Decamethonium	**−** **0.26 (0.043)**	−0.04 (0.75)	−0.06 (0.65)	**−** **0.26 (0.047)**	−0.23 (0.072)	−0.23 (0.07)	−0.24 (0.056)
Gemcadiol	**−** **0.29 (0.021)**	−0.004 (0.98)	−0.06 (0.63)	**−** **0.25 (0.049)**	−0.14 (0.29)	−0.25 (0.056)	0.24 (0.066)
Clofilium	**−** **0.28 (0.026)**	0.23 (0.079)	−0.19 (0.14)	−0.25 (0.051)	−0.14 (0.29)	**−** **0.27 (0.03)**	−0.06 (0.64)
1-[[4,5-Bis(4-methoxyphenyl)-2-thiazolyl]carbonyl]-4-methylpiperazine	−0.13 (0.33)	−0.15 (0.24)	**−** **0.29 (0.025)**	−0.17 (0.19)	**−** **0.28 (0.029)**	−0.17 (0.19)	0.04 (0.78)
Zindotrine	−0.03 (0.82)	0.17 (0.18)	0.05 (0.68)	**−** **0.28 (0.03)**	−0.19 (0.15)	0.04 (0.74)	**−** **0.31 (0.015)**
Eliglustat	−0.04 (0.78)	−0.01 (0.94)	**−** **0.26 (0.04)**	**−** **0.27 (0.036)**	−0.1 (0.44)	−0.16 (0.22)	−0.01 (0.93)
3-(Azepan-1-ylsulfonyl)-N-(3-bromophenyl)benzamide	−0.02 (0.87)	−0.25 (0.054)	**−** **0.26 (0.04)**	−0.08 (0.54)	−0.11 (0.43)	−0.08 (0.53)	**−** **0.25 (0.047)**

## Data Availability

All data supporting reported results can be found in the manuscript and Appendix A.

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
