# Peer review of "From Gene Networks to Therapeutics: A Causal Inference and Deep Learning Approach for Drug Discovery"

_pharmaceuticals, 2025, doi:10.3390/ph18091304_

Round 1
Reviewer 1 Report
Comments and Suggestions for Authors
This manuscript presents a robust and well-structured computational framework combining network biology, statistical mediation analysis, and deep learning-based compound screening to identify therapeutic targets and repurposable drugs for idiopathic pulmonary fibrosis (IPF). The methodology is comprehensive, the results are reported, and the integration of causal inference with deep learning represents a novel and timely contribution to the field of computational drug discovery.
Major:
- Biological Validation:
The study is computational and lacks in vitro or in vivo experimental validation. While this may be beyond scope, at least in silico validation against known antifibrotic drugs or CRISPR knockout datasets could further support findings. - Mediation Model Assumptions:
The authors mention linear assumptions in the CWGCNA mediation framework. However, gene-phenotype relationships can be highly non-linear. The limitations should be further elaborated, and potential non-linear mediation models could be discussed. - Deep Learning Model Generalizability:
The DeepCE model is trained on the LINCS dataset. It would be valuable to comment on how well this model performs outside this constrained dataset (e.g., primary cells or organoids), especially for fibrotic tissues. - Confounder Adjustment:
Although the authors adjusted for age and smoking status, other variables like sex, comorbidities, or batch effects could be impactful in such datasets and are not discussed in detail.
Minor Comments
Typo in "Abstract": Line 9 — "Abstrct" should be corrected.
- Figures:
Figure references (e.g., Figure 1, Figure 2, Figure 4) are useful but could benefit from more precise captions explaining what each part (a), (b), etc., represents. Some plots (e.g., volcano, forest) are described in text but the resolution or labels are not shown in this PDF version. - Supplementary Files:
The manuscript frequently refers to Supplementary Files 1–6. A brief description of the content in each of these files should be included in the main text or in a dedicated supplementary section. - Code Repository:
The authors have shared a GitHub link. A DOI (e.g., via Zenodo) would improve permanence and citability of the code. - Drug Ranking Criteria:
While compounds are selected based on negative Spearman correlation, further metrics such as toxicity profile, drug-likeness, or approval status could be briefly discussed for top candidates.
Author Response
Answers are uploaded

Reviewer 2 Report
Comments and Suggestions for Authors
Sudhir Ghandikota et al. have presented a draft on From Gene Networks to Therapeutics: A Causal Inference and 2 Deep Learning Approache for Drug Discovery. Here are some of my comments for the same.
- The authors used a power parameter (β = 8) for WGCNA without providing a detailed justification.
- The rationale for selecting the seven modules from the total sixteen is only loosely tied to p-value < 0.05. Please clarify how biological relevance was factored into this decision.
- The age and smoking status were controlled for in the mediation models; the method of adjustment and its validation should be elaborated.
- The manuscript uses CWGCNA for forward and reverse causal mediation but does not clearly describe how the directionality of causal inference was validated across cohorts.
- It is not clear how multiple hypothesis testing was addressed when computing p-values for over 2000 compounds.
- The process of selecting the 145 causal mediator genes as the phenotype signature for compound screening lacks clarity on thresholds like proportion mediated and CI ranges.
- The study employs external datasets (GSE124685 and GSE213001) for validation. Please clarify how batch effects and platform differences were normalized.
- The intersection of mediator genes with disease stages (IPF1–3) is valuable. However, presenting these results in a visual format through heatmaps or Venn diagrams would aid interpretation.
- The drugs like telaglenastat and merestinib are linked to IPF biology; the discussion lacks details on dosing, IC50 values, or prior in vivo validation in fibrosis models.
- The compound screening used seven cell lines, many of which are cancer-derived. A rationale for their selection in the context of lung fibrosis should be provided.
Author Response
Answers are uploaded

Reviewer 3 Report
Comments and Suggestions for Authors
The authors present here a bioinformatic approach that combines network analysis with deep learning based screening of compounds for drug repurposing. As case study, they analyze the idiopathic pulmonary fibrosis (IPF). They first identify genes whose expression is altered in the disease from transcriptomic data of a large number of patients and controls, then predict through statistical mediation techniques the potential causal genes, and finally they performed a DL-based screening to identify molecules potentially active on IPF.
The paper is well written and I do not have major concerns.
Lane 66 “phenotypic signature”: I would say “genotypic signature of the disease” if the authors mean that, otherwise please clarify.
Par. 2.3 : Candidate causal genes for IPF. This is part should be improved: the assumption to infer causal genes, responsible of the disease, from expression only, is an ambitious goal. Data should be compared with genetic association (available in many websites such as opentarget) and differences should be discussed.
Par.2.4 Not clear if authors are looking for disease drivers or markers here: They start with : “To identify potential disease-driving mediator candidates…”, but they end the paragraph with “In 147 total, our mediation analysis identified 54 unique candidates that are marker genes in at 148 least one of the three pro-fibrotic niches”. Please clarify.
In general in the paper markers and causal genes should be properly differentiated and addressed when describing the results.
Lane 232: “after adjusting for confounders” is definitely not clear here.
Lane 316: again what author mean with “confounding variables”? Patientis characteristics were associated to each rna-seq or transcriptomic file?. Please specify what authors have observed to consider some variables confounding and others not.
Author Response
Answers are uploaded

Reviewer 4 Report
Comments and Suggestions for Authors
This manuscript presents a computational framework integrating weighted gene co-expression network analysis (WGCNA), bidirectional mediation analysis, and a deep learning–based compound screening platform (DeepCE) for drug discovery and repurposing, applied to idiopathic pulmonary fibrosis (IPF). Several points require clarification, additional analysis, or improved presentation before the manuscript is suitable for publication.
The Introduction could better position the work within the context of prior studies that have applied network-based and AI-driven repurposing methods for complex diseases. For example, how does this framework outperform or complement approaches such as direct differential expression–based connectivity mapping or other causal network inference pipelines?
The biological interpretation of the top genes (ITM2C, PRTFDC1, CRABP2, CPNE7, NMNAT2) could be expanded. Are these genes known to be involved in fibrosis, extracellular matrix remodeling, or immune dysregulation in lung tissue?
Line 127: Please clarify what is meant by the “druggable genome collection,” including references to the source and the criteria used for defining this collection.
The Methods and Results sections are not assigned appropriately. Figures should be cited primarily in the Results section to clearly indicate to readers what findings each figure illustrates.
It would be beneficial to include an additional figure showing how the genes are clustered and how the different modules are distributed within the gene network.
Section 4.3 is unclear. Machine learning models are typically built for prediction purposes; the authors should explicitly state what the inputs to these models are and what outcomes or variables the models are intended to predict.
Line 9: In the Abstract, “Abstrct” is misspelled and should be corrected.
Line 114: The reference to “Figure 1” appears to be incorrect and should be “Figure 2.”
Figures 1–4 contain dense information; adding clearer legends and more descriptive captions would improve readability.
Author Response
Answers are uploaded

Reviewer 5 Report
Comments and Suggestions for Authors
The manuscript presents a novel and comprehensive computational framework that integrates network analysis, causal inference, and deep learning for drug discovery, with a focus on idiopathic pulmonary fibrosis (IPF). The study is well-designed and addresses an important gap in the field by identifying causal genes and potential therapeutic candidates. The methodology is robust, and the results are promising. The manuscript presents a significant contribution to the field of computational drug discovery, particularly for complex diseases like IPF. With major revisions to address the following points, it would be suitable for publication.
Consider briefly mentioning the limitations of existing approaches (e.g., single-target paradigms) earlier in the introduction to better contextualize the novelty of the proposed framework.
The description of the WGCNA and mediation analysis is thorough, but some technical details (e.g., parameter selection for WGCNA, adjustments for confounders in mediation analysis) could be expanded for reproducibility.
Clarify how the DeepCE model was customized for this study, particularly the rationale behind using BING genes instead of landmark genes.
Provide more details on the validation cohorts (e.g., sample characteristics, how they were selected) to ensure readers understand the generalizability of the results.
The manuscript would benefit from a deeper discussion of the biological relevance of the top candidate genes (e.g., ITM2C, CRABP2) and their potential roles in IPF pathogenesis.
The rationale for selecting the top candidates (e.g., Telaglenastat, Merestinib) could be strengthened by linking their mechanisms of action more explicitly to IPF pathways.
Include a summary table or figure that consolidates the key findings (e.g., causal genes, biomarker performance, top compounds) for easier interpretation.
In discussion, highlight the strengths of the framework: For example, how might unmeasured confounders or non-linear relationships impact the mediation analysis?
Elaborate on the translational potential of the identified compounds. Are there any known safety or efficacy data for these drugs in IPF or related fibrotic diseases?
Consider adding a schematic to illustrate the biological pathways or networks involving the top causal genes and compounds.
Some acronyms (e.g., BING, GLM) are defined late in the text or in supplementary materials. Define them upon first use for clarity.
The manuscript is well-written but would benefit from a thorough proofread to correct minor grammatical errors (e.g., "IFF" instead of "IPF" on page 10).
Author Response
Answers are uploaded

Round 2
Reviewer 2 Report
Comments and Suggestions for Authors
The authors have addressed the assigned comments at a satisfactory level.
Reviewer 4 Report
Comments and Suggestions for Authors
My comments have been addressed by the authors